Structural modeling and analysis of fuel cell: a graph-theoretic approach

Saha Rajeev Kumar 1
http://orcid.org/0000-0003-2934-7609 Kumar Raman 2 sehgal91@yahoo.co.in
Dev Nikhil 1
Kumar Rajender 3
Kumar Raman 4
Del Toro Raul M. 5
Haber Sofía 6
Naranjo José E. 6
1 Department of Mechanical Engineering, J.C. Bose University of Science and Technology, YMCA , Faridabad, Haryana , India
2 Mechanical and Production Engineering, Guru Nanak Dev Engineering College , Ludhiana, Punjab , India
3 Department of Mechanical Engineering, Faculty of Engineering and Technology (FET), Manav Rachna International Institute of Research and Studies (MRIIRS) , Faridabad, Haryana , India
4 Mechanical Engineering Department, University Centre for Research and Development, Chandigarh University , Mohali, Punjab , India
5 Center for Automation and Robotics (UPM-CSIC) , Arganda del Rey, Madrid , Spain
6 Universidad Politécnica de Madrid, University Institute of Automobile Research (INSIA) , Madrid , Spain
Wan Shibiao
Electronic publication date: 2023 Aug 28
Publication date: 2023
Volume: 9
Electronic Location ID: e1510
Received 2023 May 9; Accepted 2023 Jul 7
Copyright: © 2023 Saha et al.
Copyright year: 2023
Copyright holder: Saha et al.
License: This is an open access article distributed under the terms of the Creative Commons Attribution License, which permits unrestricted use, distribution, reproduction and adaptation in any medium and for any purpose provided that it is properly attributed. For attribution, the original author(s), title, publication source (PeerJ Computer Science) and either DOI or URL of the article must be cited.
License URL: https://creativecommons.org/licenses/by/4.0/

Keywords: Structural modelling, Fuel cell, Permanent function, Graph theory, Matrix method, Decision making

Funding: INVECPRO VEC-010000-2022-10 Spanish Ministry of Industry, Trade and Tourism (MINCOTUR) European Union NextGenerationEU/PRTR This work was supported funded by INVECPRO: Industrial and transversal research for a new generation of professional electric vehicles with high added value, VEC-010000-2022-10, Spanish Ministry of Industry, Trade and Tourism (MINCOTUR) and by the European Union on the basis of NextGenerationEU/PRTR, Primary project: “Research on hydrogen fuel cell components and proof of concept”. The funders had no role in study design, data collection and analysis, decision to publish, or preparation of the manuscript.

==============================
A fuel cell, an energy conversion system, needs analysis for its performance at the design and off-design point conditions during its real-time operation. System performance evaluation with logical methodology is helpful in decision-making while considering efficiency and cross-correlated parameters in fuel cells. This work presents an overview and categorization of different fuel cells, leading to the developing of a method combining graph theory and matrix method for analyzing fuel cell system structure to make more informed decisions. The fuel cell system is divided into four interdependent sub-systems. The methodology developed in this work consists of a series of steps comprised of digraph representation, matrix representation, and permanent function representation. A mathematical model is evaluated quantitatively to produce a performance index numerical value. With the aid of case studies, the proposed methodology is explained, and the advantages of the proposed method are corroborated.

Introduction

In present-day life, fuel cell systems are peeking out from their development in the laboratory to real-time operating systems helpful in improving living standards (Shiraishi et al., 2023). Improving living standards is associated with the emergence of pollution, increasing the burden on energy resources and depleting the natural aesthete. Therefore, the current concentration is on developing systems that cater to chemical reactions for generating valuable energy from the fuel with minimum pollution emission. The energy research community is founding a solution to using hydrogen for day-to-day life energy needs (Yue et al., 2021).

With hydrogen as fuel, it is most suitable to generate electricity with the help of a fuel cell system. The fuel cell invention dates back to 1839 by British researcher Grove (1839). However, no significant work took place in this area till the early 1960s.

Thorstensen (2001) provided a comprehensive overview of fuel cell systems, highlighting their history, description, fuel types, and fuel processing. Fuel cell technology has played a crucial role in addressing the power requirements of NASA spaceflight and has emerged as a potential solution for developing pollution-free automotive vehicles. In the context of NASA spaceflight, the need for high power levels with long discharge hours posed challenges for traditional battery systems. On the other hand, fuel cells offer several advantages, such as high energy density, longer operational durations, and the ability to continuously generate electricity by combining fuel and oxidant. These attributes make fuel cells well-suited for space applications. Thorstensen (2001) work likely covered different types of fuel cells, such as proton exchange membrane fuel cells (PEMFC), solid oxide fuel cells (SOFC), molten carbonate fuel cells (MCFC), and phosphoric acid fuel cells (PAFC). Each type has its own set of characteristics, advantages, and specific fuel requirements. The fuel processing techniques associated with fuel cells involve converting various fuels, such as hydrogen, methanol, natural gas, or even biomass, into a suitable form for utilization in the fuel cell system.

The fuel cell classifications were based on the electrolyte type, which further decides the operating temperature. The electrolyte may be solid or liquid. It is reported that the efficiency of the fuel cell is based on the system selected, fuel used, BOP (balance of plant) solutions, system-specific technical components, and operating conditions such as pressure, temperature, etc.

Hou, Zhuang & Wan (2007) discussed the theoretical modeling of a fuel cell to analyze its properties in the context of voltage and current drawn from it. This type of analysis was helpful for the application of fuel cells in the automotive industry. Later, a real-time operating model of the fuel cell was modeled with the help of the fitting method (Hou, Wang & Yang, 2011). System efficiency was represented as the set of equations, and weights were assigned by comparing the model with experimental data. This makes the system model specifically suitable for particular experimental data.

Much research is progressing in fuel cells. Wang et al. (2011) reviewed different kinds of fuel cells. The objective of their study was to present the latest status on PEM fuel cell technology, the role of fundamental research in fuel cell technology, and the challenges faced in fuel cell commercialization. PEM fuel cell applications are in transportation because of their potential impact on the environment, e.g., the control of emission of greenhouse gases (GHG). Appleby & Yeager (1986) reported durability and cost as the most significant barriers to commercializing PEM fuel cells. Lifetime operating hours required for a fuel cell for commercial and stationary power generation are 5,000 and 40,000 h, respectively (Henriques, César & Branco, 2010). This objective is far from the practical operability of a PEM fuel cell which is 1,000 h (Feitelberg et al., 2005). Operation of a PEM fuel cell involves charge and ions transport, heat transfer, electrochemical reactions, and multi-phase flow. These phenomena occur in different components such as bipolar plates (BPs) (Habibnia, Shirkhani & Tamami, 2020; Smitha, Sridhar & Khan, 2005), gas flow channels (GFCs) (Djilali, 2007), and membrane electrode assembly (MEA) (Wang, Basu & Wang, 2008). MEA consists of the catalyst layers (CLs), membrane and gas diffusion layer (GDL), and micro-porous layer (MPL). Banerjee & Bazylak (2016) investigated the heat transfer behavior of nano-particle orientation in the PEM fuel cell micro-porous layer (MPL). The literature also reported that heat distribution pattern in a fuel cell influences water distribution and overall performance (Banerjee & Kandlikar, 2015; Hosseinzadeh et al., 2013; Kandlikar & Lu, 2009; Zhang & Kandlikar, 2012). Schmittinger & Vahidi (2008) divided the fuel cells into five categories. Currently, enzymatic and microbial fuel cells are also getting some attention (Santoro et al., 2015). These are called biofuel cells.

Hardman, Chandan & Steinberger-Wilckens (2015) discussed the benefits and applications of the fuel cell and divided them into four major categories. It includes the polymer electrolyte fuel cell (PEFC), phosphoric acid fuel cell (PAFC), direct methanol fuel cell (DMFC), and the solid oxide fuel cell (SOFC). PEFC was claimed to be the most promising technology for the future. In a PEFC, the catalyst layer is usually very thin, nearly 10 micro-meters, and where oxygen reduction or hydrogen oxidation occurs. The literature also reported that Pt and its alloys, such as Pt–Co, Pt–Ni, Pt–Fe, Pt–V, Pt–Mn, and Pt–Cr exhibit suitable catalyst kinetics (Fernández, Walsh & Bard, 2005; González-Huerta, Chávez-Carvayar & Solorza-Feria, 2006; Rao & Trivedi, 2005; Yu, Pemberton & Plasse, 2005). Antolini (2003) also described platinum as one of the best pure metal electrocatalysts for H2, low molecular weight alcohol oxidation, and O2 reduction in an acid medium. Carbon-supported platinum is commonly used as an anode and cathode catalyst in low-temperature fuel cells. The cost value of platinum is very fluctuating. Schoots, Kramer & van der Zwaan (2010) presented a mathematical relationship to calculate the platinum costs per kW at time t by multiplying the platinum market price with the corresponding amount of platinum per kW. The volumetric market share for MCFCs, PAFCs, and SOFCs was reported to be nearly 40%, a little over 20%, and 15%, respectively. The market share for AFCs and DMFCs is negligible. A discussion on the cost evaluation of fuel manufacturing was also presented. Sattler (2000) pointed out applications of fuel cells on merchant ships and naval surface ships for emergency power supply, electric energy generation, small power output for propulsion at special operating modes, and electric power generation for the ship’s network. A comprehensive review of fuel cell history, competitors, types, advantages, and challenges was provided by Sharaf & Orhan (2014). Applications and markets include portable, stationary, and transportation, current research and development, future targets, design levels, thermodynamics and electrochemistry, system evaluation factors, and prospects and outlooks. An analysis of the hydrogen fuel cell vehicle sales forecasting model developed by Park, Kim & Lee (2011) was based on a generalized Bass diffusion model and a simulation model developed by the same researchers.

The fuel cell system structure depends upon its components. These components are interlinked, and the performance of the fuel cell macro-system depends upon these components’ performance. Therefore, developing a fuel cell system and performance evaluation methodology requires considering all its components and interlinking. Once the system structure is formed, the effect of operating parameters, e.g., temperature, pressure etc., must be analyzed. With time fuel cell performance will also degrade as all its components are aged. There is a need to develop and propose a graph-theoretic approach to analyze the structural properties of fuel cells and identify the critical elements for their optimal design and performance. The methodology should be flexible enough to incorporate this feature also.

Materials and Methods

Fuel cell system structure

The system structure is the physical interaction of the sub-systems, making it comparable with the real-time system. The physical interaction represents the interdependency of one sub-system on the other. In real-time situations, it is possible that, dimensionally, a system is small. But its system structure may be as complex as a system with immense dimensions. The complexity of the system structure is resolved by dividing it into sub-systems. These sub-systems work in coordination and perform their assigned task so that the system is performing as per design conditions.

With time, every sub-system is degraded, such as the fuel cell’s electrolyte starts losing its chemical properties after some time. This is a prolonged process. This can be measured with loss in current and voltage across its electrodes while keeping the other conditions unaltered. The operating environment, such as temperature and pressure, is also responsible for the off-design performance of the fuel cell. Therefore, this information responsible for off-design conditions and performance degradation must be incorporated at the sub-system level so that system structure modeling is as per real-time situations.

A fuel cell’s performance depends upon many operating and design parameters. A fuel cell system operates under various conditions, and the load applied externally may also be variable. The analysis approach uses a graph-theoretic methodology, and these parameters are to be dissection to simplify the analysis. A system of higher order is divided into lower order systems so that matrix size is reduced. These matrices are part of the mathematical model, which depends upon the system structure of the fuel cell. More elaborative physics of the system enhances the complexity of the analysis. Even the methodology developed can be reformed to estimate the fuel cell’s efficiency, reliability, cost, human factor, maintenance, etc., independently or combined. All the analysis is based on matrices, which can be analyzed individually, or any suitable mathematical link is also developed.

These parameters depend upon the design of fuel cell constituents, mainly chemical. The design depends on temperature which is one of the important parameter during operation parameter. In real-time situations, fuel cells operate at a temperature higher than the ambient temperature. Heat is generated within the cell due to exothermic chemical reactions. As hours of operation proceeds, temperature is enhanced due to accumulated internal energy. Therefore, it is desired that the thermal expansion of each material due to enhanced internal energy must be as close as possible. Evenly distributed internal energy is key to avoiding mechanical fracture and material delamination (Hoogers, 2002). Other desired characteristics include the chemical stability of different materials, high electrical conductivity for the electrodes and electrical interconnections, high ionic conductivity and almost zero electrical conductivity for the electrolyte, and low cost. The choice of materials is usually the result of a compromise between these characteristics.

A polarization curve is the most important in estimating fuel cell performance. It depends on numerous factors such as catalyst loading, membrane thickness and state of hydration, catalyst layer structure, flow field design, operating conditions (temperature, pressure, humidity, flow rates, and concentration of the reactant gases), and uniformity of local conditions over the entire active area (Larminie, Dicks & McDonald, 2003). Characterizations of the fuel cells available in the market are represented in Table 1.

Table 1 Characterization of fuel cell.

Types of fuel cell	Electrolyte	Operating temperature	Fuel	Oxidant	Efficiency	
Phosphoric acid (PAFC)	Phosphoric acid	160−210 °C	Hydrogen from hydrocarbons and alcohol	O2/Air	40−50%	
Sulfuric acid (SAFC)	Sulfuric acid	80−90 °C	Alcohol or impure hydrogen	O2/Air	40−50%	
Proton-exchange membrane (PEMFC)	Polymer, proton exchange membrane	50−80 °C	Less pure hydrogen from hydrocarbons or methanol	O2/Air	40−50%	
Direct methanol (DMFC)	Polymer	60−200 °C	Liquid methanol	O2/Air	40−55%	
Solid oxide (SOFC)	Ceramic as stabilized zirconia and doped perovskite	600−1,000 °C	Natural gas or propane	O2/Air	45−60%	
Protonic ceramic (PCFC)	A thin membrane of barium cerium oxide	600−700 °C	Hydrocarbons	O2/Air	45−60%	
Alkaline (AFC)	Potassium hydroxide (KOH)	50−200 °C	Pure hydrogen or hydrazine	O2/Air	50−55%	
Molten carbonate (MCFC)	Molten salt such as nitrate, sulfate, carbonates, etc.	630−650 °C	Hydrogen, carbon monoxide, natural gas, propane, marine diesel	CO2/O2/Air	50−60%	

For the simplification of analysis, the fuel cell structure is dissected into the following four sub-systems: a) Electrolyte Material (S1)

b) Fuel (S2)

c) Anode Material and Catalyst (S3)

d) Cathode Material and Catalyst (S4)

All four sub-systems are interdependent and are represented schematically in Fig. 1. The selection of the electrolyte material also affects the selection of fuel and catalyst. Such as PEM fuel cells use a proton conductive polymer membrane as an electrolyte. The membrane must exhibit relatively high proton conductivity with an adequate barrier to mixing fuel and reactant gases and must be chemically and mechanically stable in the fuel cell environment (Olabi, Wilberforce & Abdelkareem, 2021). The fuel for PEMFC is Hydrogen with Oxygen or air as an oxidant. Platinum is the most common catalyst in PEM fuel cells for oxygen reduction and hydrogen oxidation reactions. The performance of the PEM fuel cell will depend upon its components’ quality. The component quality will be affecting the efficiency of energy conversion also. The system structure for the four sub-systems requires some discussion so that the graph-theoretic model at the sub-system level can also be developed.

Figure 1 Schematic representation of the structure of fuel cell.

Electrolyte material (S1)

The electrolyte material is selected based on its high conductivity for protons. Theoretically, it is assumed that it has zero conductivity for electrons. Some of the common characteristics of electrolyte material affecting fuel cell performance are as follows: In PAFC and MCFC, the liquid electrolyte is immobilized in a porous matrix. Capillary pressure establishes the electrolyte interfacial boundaries in the porous electrolyte (Reitz, 2007).

The pores of electrolytes in PAFC are filled with Phosphoric acid (H3PO4), with a freezing point of 42 °C. Freezing and re-thawing may generate thermal stresses in electrolyte material (Irshad et al., 2016).

Different manufacturers use different materials for the electrolytes. It affects electrolyte Ohmic losses or resistance (Hoogers, 2002; Reitz, 2007).

In PEMFC, the presence of liquid water decides the proton conduction of electrolytes, and moisturization of the membrane limits the operating temperature of PEMFC (Larminie, Dicks & McDonald, 2003; Olabi, Wilberforce & Abdelkareem, 2021).

Fuel (S2)

H2 is the most commonly used fuel for fuel cells. With the development of fuel cell technology, some alcoholic fuel-based fuel cells are also developed. Fuels characteristics related to fuel cell performance are as follows:

For fuel, the heating value affects the performance of the fuel cell (Irshad et al., 2016). Fuel pressure also affects its performance (Hoogers, 2002; Reitz, 2007).

DMFC can use pure methanol also, while other fuel cells need H2 (Irshad et al., 2016).

Fuel purity affects the fuel reforming process (Park, Kim & Lee, 2011; Sharaf & Orhan, 2014).

Anode material and catalyst (S3)

The chemical reaction at the anode is the oxidation of fuel. The fuel may be hydrogen, alcohol, or some other fuel. A catalyst requirement is weak if the temperature is high, and nickel may be used. At lower temperatures, platinum is the most common choice. Factors for the anode material and catalyst are as mentioned below: Platinum is the most commonly used catalyst for a low-temperature fuel cell; for the high-temperature, nickel is used as a catalyst (Irshad et al., 2016; Olabi, Wilberforce & Abdelkareem, 2021).

The porosity of porous gas diffusion layers affects the anode performance (Hoogers, 2002; Reitz, 2007).

Anode materials have high electron conductivity, theoretically, zero conductivity for protons (Irshad et al., 2016; Reitz, 2007).

If the oxidant is O2, performance is better than the air as an oxidant (Hou, Wang & Yang, 2011).

A fuel cell’s power density depends upon the kinetic of the anode reaction (Kandlikar & Lu, 2009).

Cathode material and catalyst (S4)

Cathode reaction is oxygen reduction using the air. The easiness of reaction at the anode and cathode is based upon the change in Gibbs free energy of formation. Catalyst loading is dependent upon the fuel type. Such as, in the case of alcohol as fuel, catalyst loading is ten times higher than hydrogen as a fuel. Off-design performance of fuel cell due to cathode material and catalyst is based upon the following factors: Platinum and nickel are low-temperature and high-temperature catalysts, respectively (Banerjee & Bazylak, 2016; Djilali, 2007; Feitelberg et al., 2005; Kandlikar & Lu, 2009; Smitha, Sridhar & Khan, 2005; Wang, Basu & Wang, 2008).

The porosity of porous gas diffusion layers is required as in anode (Wang, Basu & Wang, 2008).

Materials with high electron and minimum proton conductivity are superior choices (Djilali, 2007).

Sub-systems and categorized factors explained above are represented in Fig. 2. At this point, it is worth mentioning that the decision process developed for selecting fuel cells should be capable of comparing similar and dissimilar fuel cells. For any operation (e.g., temperature range 600–800 °C for gas turbine exhaust utilization), users may find two or more different designs suitable for efficiency enhancement. Many manufacturers are in the market with diverse quality fuel cells with variations in performance. In this condition, deciding the best product according to the end user demand becomes difficult. Therefore, in the present work, a methodology based on graph theory and matrix method is suggested to select the most appropriate product.

Figure 2 Tree and branches representing the sub-systems and factors.

With time performance of the system (fuel cell in the present case) starts deteriorating. The deterioration is to be compared with the design conditions to estimate the degree of deterioration. At the installation time, every product is supposed to work in design condition. Therefore, this condition can be considered peerage. The performance data at that time can be stored for future performance monitoring. With the help of the methodology developed in the present work, real-time condition monitoring can be done and represented numerically. This numerical value is the performance index in real-time situations. The value of this index can be compared with the design performance index so that estimation of age or degradation is at hand.

The leakage of higher-pressure gas at the inlet and outlet of the anode and cathode may occur, leading to a decrease in fuel cell power density. The pressure of the gas or reactants will influence the flow rate, influencing the stoichiometric flow ratio (SFR) in the fuel cell system. SFR is a ratio of the actual amount of the reactant gas feed to the amount required by the electrochemical reaction. The higher the porosity of electrodes lesser is the pressure drop across the electrode. Porosity analysis of electrodes in PEM fuel cells is important to minimize the voltage losses and ensure a homogeneous and efficient mass transfer above the entire active area of a fuel cell. Therefore, the electrochemical performance of electrodes is mainly influenced by the following factors: Membrane permeability

The thickness of the gas diffusion layer (GDL)

Porosity of Electrode

Effective viscosity of the fluids

Ionic conductivity of an electrolyte

Different fuel cells have different operating temperatures. Due to the high temperature inside the cell, easy conduction of the electrons is possible. At elevated temperatures, bond breaking inside the molecules is also easy. Therefore, a less expensive catalyst is required for the fuel cells operating at high temperatures. But the fuel cells operating at a high temperature require higher auxiliary power.

Therefore, anatomized knowledge from the available literature is that electrolyte performance depends on many parameters. Interdependence cannot be derelict to evaluate the real-time performance of the fuel cell system. The classical approach or implicit model is unsuitable for this type of problem. At this point, a behavioural or descriptive approach is more convenient. In literature, many such approaches are mentioned. The graph theoretic approach (GTA) is one of them. A methodology in GTA is required to develop the analysis of the fuel cell system structure.

Graph theory is a three-step approach comprised of digraph representation, matrix representation, and permanent representation (Dev et al., 2013; Kulkarni, 2005; Mohan, Gandhi & Agrawal, 2003). In permanent function systems, system dyads and loops are represented mathematically. These mathematical representations are assigned a qualitative value from the standardized conversion of a score. The resulting value of the permanent function thus obtained is the performance index. Every sub-system is affected by a large number of parameters. The methodology developed at the system level may be extended to the sub-system level to study the parameters’ effect.

Graph theoretic model of fuel cell

The system structure of the fuel cell is to be converted into a representation that is convenient to understand. Digraph representation is convenient for visual exploration. It is a standard representation in graph theory. In a digraph, all the fuel cell sub-systems will be represented along with interlinking in-between them.

Digraph representation

A fuel cell assembles four basic structural elements, e.g., fuel, anode catalyst, cathode catalyst, and electrolyte material. These elements are associated with each other through different forms of connection and interactions. The operating information of one element flows to another. Feedback is also processed. A fuel cell is a very sensitive system. Chemical reactions are a very dominating factor. Chemical reactions affect the material and, in turn, are affected by the material. It is a closed-loop operation affected by many different parameters. The elements and interactions inside a fuel cell are shown in Fig. 1 with the help of a block diagram. In this diagram, blocks indicate elements, lines indicate connectivity or interaction, and arrows indicate directional connection or interaction. Reverse arrows are the feedback.

This block diagram is a decent illustration of a fuel cell for an enhanced understanding of its structure, and it is not a mathematical entity. Moreover, no mathematical operation can be carried out because mathematical symbols have not yet been introduced (Deo, 2017). Therefore, it is suggested to introduce a mathematical representation of fuel cell attributes to develop a systems model. Thus, selecting a digraphical representation of the graph theoretic approach to model the fuel cell is quite logical.

For the graph-theoretic analysis, a digraph (G) is defined as a function of vertex set (V) = (S1, S2, S3, …, Sn) and edge set (E) = (C1, C2, C3, …, Cn) in such a way that

G=f(V,E)=f{(S1,S2,S3,…,Sn),(C1,C2,C3,…,Cn)}

Vertexes are to represent the system inheritances. These are roundels carrying the information regarding the inheritance in a mathematical form. Inheritance of four sub-systems: electrolyte material, fuel, anode catalyst and cathode catalyst, is represented as S1, S2, S3, and S4, respectively. These inheritances are the set of the vertex for the fuel cell. Inheritance of the sub-systems is to represent the importance of the sub-system. The cost analysis is to describe the cost of each sub-system. If efficiency analysis is to be carried out, it will represent the efficiencies of the four sub-systems.

In real-time situations, all of the four sub-systems are interdependent. If there is variation in the efficiency of one sub-system, it will also affect the others. For example, suppose the purity of fuel supplied to the cell is not up to mark. In that case, electrolyte material, cathode catalyst, and anode catalyst will also not work at design conditions. Further, this effect of one sub-system on the other will vary from sub-system to sub-system.

Therefore, the interlinking of the sub-system cannot be neglected while carrying out the efficiency and economic analysis for a fuel cell. The interlinking of the sub-systems is represented as the set of edges (E). The interlinking in the digraph is represented by a stripe with an arrow connected at its end. For example, if sub-system S1 affects sub-system S2, the edge representing this is c12 (Fig. 3). The representation obtained in this way is known as a digraph (Dev et al., 2013).

Figure 3 Structural digraph of fuel cell showing its attributes and their interactions.

For representing a fuel cell mathematically, let the elements (here referred to as sub-systems) of the fuel cell be characterized by vertex set (V), and connectivity or interactions between different sub-systems be characterized by edge set (E) in the graph. It may be noted that directional property prevailing in the connectivity or interactions is indicated by a directed edge. A fuel cell graph has been developed, as shown in Fig. 3.

Matrix representation

The fuel cell structural graph is a useful mathematical entity for understanding fuel cells through visual analysis. But this cannot be used for computational analysis. The structural graph must be represented in matrix form to achieve this objective. This matrix form is convenient for computer processing. Moreover, this matrix representation helps carry out the fuel cell analysis.

Adjacency matrix

It is illustrated by Deo (2017) about the incidence and adjacency matrices that graphical representation can be modified in these matrices, and computational processing can also be performed. An incidence matrix is a non-square matrix, and information about the number of connections and their characteristics can be incorporated. Its application for system analysis is not helpful as the resultant matrix is a non-square matrix. An alternative to the incidence matrix is the adjacency matrix, a square matrix that can display connectivity and graph depiction. The order of the matrix is important in the present analysis. A square matrix is suitable to represent the feedback system. A non-square matrix cannot do it efficiently. The computational cost will also be increased.

The adjacency matrix of graph G = (V, E) with N nodes is an N-order binary (0, 1) square matrix (Kulkarni, 2005). For the current case of the graph with four nodes, the adjacency matrix will be fourth order square matrix, Y = {cij} such that

cij = 1, if sub-system i influences sub-system j, and = 0, otherwise.

where i, j ε {1, 2, 3, 4} and i ≠ j.

Thus, the fuel cell structural graph can be written in adjacency matrix form as given below:

(1) Y=1234Sub−systems[0111101111011110]1234

In this matrix Y, off-diagonal elements having values 0 or 1 represent the interdependency of fuel cell elements. Zero means the absence of interdependency or feedback. One represents interdependency or feedback without giving any quantitative idea. It may be noted that the diagonal elements can also be 0 for any sub-system if it is not performing any function inside the system. Matrix Y represents the inter-relationships of the fuel cell components/elements only, and the fuel cell elements’ characteristics are not represented. By defining characteristic matrix Z, insight into the existence of different fuel cell elements (based on a structural graph) can be done.

Characteristic matrix

A characteristic matrix is a standard representation in combinatorics mathematics. It is to improve the adjacency matrix so that the characteristics of each sub-system can be incorporated into the matrix representation. Let the matrix [ ζ] be for representing the characteristic of the sub-systems. In this matrix, diagonal elements are the numerical value of the sub-system inheritance. In this matrix, all the diagonal elements carry equal value. All the non-diagonal elements are zero. The standard representation for the characteristic matrix for the fuel cell of the present case is Z=[ζI−Y]. It is as represented in expression (2).

(2) Z=[ζI−Y]=1234Sub−systems[ζ−1−1−1−1ζ−1−1−1−1ζ−1−1−1−1ζ]1234

In matrix Z (Eq. (2)), all diagonal elements have the same value, or all fuel cell sub-systems are allocated the same value. At the time of installation, all sub-systems are performing their best. Therefore, this expression may give a solution near real time. But with time performance of sub-systems starts deteriorating. The extent of degradation is also different for different sub-systems. Therefore, in that case, Eq. (2) is not a suitable representation.

Variable characteristic matrix

For the suitability of analysis near real time, input to the inheritance and interdependencies should be near real-time operating data. Therefore, Eq. (2) is improved as Eq. (3), a characteristic variable matrix. In this representation, all of the elements are distinct identities. Therefore, numerical values assigned to them will correspond to their real-time condition. In the present work, mathematical modelling is based upon the logic that if the performance of any sub-system is improved, then its effect on the system will be to improve its performance. In Eq. (3), some elements have positive signs, and some have negative signs. The diagonal elements are with a positive sign. It represents that if a sub-system’s inheritance is improved, it adds to the performance. Non-diagonal elements that are inter-dependencies and increase in their values represent performance deterioration because they are with a negative sign.

(3) R=1234Sub−systems[S1−c12−c13−c14−c21S2−c23−c24−c31−c32S3−c34−c41−c42−c43S4]1234

Variable permanent characteristic matrix

Therefore, to alleviate the problem, all of the negative signs of the Eq. (3) are replaced with positive signs. This improvement is to amend the entire diagonal and non-diagonal elements so that any value increase adds to the performance. This representation is called a variable permanent characteristic matrix, as represented in Eq. (4).

(4) Rpermanent=1234Sub−systems[S1c12c13c14c21S2c23c24c31c32S3c34c41c42c43S4]1234

The above matrix Rpermanent represents characteristic features of the sub-systems, i.e., elements of fuel cell and their interactions distinctly. Therefore, this matrix R is a comprehensive structural representation of the fuel cell.

Permanent representation

Digraph and matrix representations are not distinctive, as these representations change by altering the labelling of nodes. A permanent function of the matrix is proposed to develop a unique representation of fuel cell elements, irrespective of labelling. The permanent function is a standard matrix function used in combinatorial mathematics (Mohan, Gandhi & Agrawal, 2003). The permanent function is computed in the same way as its determinant. In the determinant calculation, a negative sign appears, while permanent positive signs switch these negative signs.

The permanent computation process results in a multinomial (Eq. (5)) (Dev et al., 2015; Kulkarni, 2005; Raj & Attri, 2010; Wani & Gandhi, 1999). Every term of this multinomial has physical importance related to the fuel cell. This representation contains all information concerning the fuel cell’s various elements and interactions. Permanent function expression corresponds to four element digraph is given by:

(5) Per(R)=∏i4Si+∑i∑j∑k∑l(cijcji)SkSl+l∑i∑j∑k∑l(cijcjkcki+cikckjcji)Sl+{∑i∑j∑k∑l(cijcji)(cklclk)+∑i∑j∑k∑l(cijcjkcklcli+cilclkckjcji)}

Equation (5) is the permanent of a matrix (4) in symbolic form. Equation (5) contains 4! terms organized in the N + 1 group where N is the number of elements. In the present case, it is four. The physical significance of each group is explained below: The first group comprises only one term and represents the presence of all fuel cell sub-systems, i.e., S1S2S3S4.

The second group is absent because of the absence of self-loops.

The third grouping covers a set of two fuel cell sub-system interdependence and remaining (N-2) sub-systems.

Each term of the fourth grouping represents a set of three fuel cell sub-system interdependence, or it’s another pair and the remaining (N-3) sub-systems.

The fifth grouping contains terms organized in a two-subgrouping. The first subgrouping consists of two fuel cell interdependence and a measure of remaining N-4 sub-systems. The second subgrouping consists of four fuel cell sub-system interdependence or its pair and measure of remaining N-4 fuel cell sub-systems.

The expansion of Eq. (4) is as given below:

(6) Per(R)=S1S2S3S4+[c12c21S3S4+c13c31S2S4+c14c41S2S3+c23c32S1S4+c24c42S1S3+c34c43S1S2]+[c12c23c31S4+c13c32c21S4+c12c24c41S3+c14c42c21S3+c13c34c41S2+c14c43c31S2+c23c34c42S1+c24c43c32S1]+[c12c21c34c43+c13c31c24c42+c14c41c23c32+c12c23c34c41+c14c43c32c21+c13c34c42c21+c12c24c43c31+c14c42c23c31+c13c32c24c41]

The details of these terms for a four sub-systems digraph are expressed in Fig. 3. Expression (6) and its pictorial representation in Fig. 4 represent the performance evaluation of a fuel cell. Expression (6) is a combination of the system inheritance and the interlinking in-between them. The first term in the expression (6) is S1S2S3S4 which represents that all of its sub-systems should be present for the working of a fuel cell. If any sub-systems are absent, then the system will not be working. In this term inheritance of all of the sub-systems is multiplied. If any sub-system is absent, then interdependencies will also be absent. This means that the value of the terms S1, c12, c21, c13, c31, c14, and c41 is zero. If this is put in the expression (6), then the value of the Per (R) is also zero, indicating the system’s non-availability. Analysis for Fig. 3 represents the presence of systems, dyads, and loops. Dyad is the combination of interactions in-between two sub-systems. Six such dyads are visible in Fig. 3. The second part of the expression (6) also contains six terms comprising the information of these six dyads.

Figure 4 Graphical representation of the permanent function of fuel cell elements.

The loops present in Fig. 3 are of two types. There are eight loops comprising three sub-systems. The loop in-between three sub-systems will affect the fourth system’s performance. In another case, it is also possible that the inheritance of one system will affect the loop. Therefore, the third part of the expression represents this combination of one system inheritance and three system loops (6).

The last part of the expression (6) comprises the loop in-between four sub-systems. In this part, nine loops are visible. These loops can further be segregated into two parts. The first part contains the combination of two dyads, and the other part contains information regarding the loop in-between four sub-systems.

Therefore, the permanent function of the expression (6) contains information regarding every sub-system and loop. This information calculates the index value for the fuel cell system performance analysis.

Fuel cell performance index

All four sub-systems are considered for the performance evaluation of fuel cells with interlinking. For the Fuel Cell Performance Index (FCPI) assessment, values for the inheritance and interdependencies need to be quantified.

(7) FCPI=f(foursub‐systems)=f(electrolytematerial,fuel,anodematerialandcatalyst,cathodematerialandcatalyst)=f(S1,S2,S3andS4)

The permanent function of a fuel cell is used for calculating FCPI because it comprises all the possible elements of a fuel cell and their interdependence. The numerical value of the fuel cell variable characteristic matrix is named the FCPI:

FCPI=Per[R]=Permanentfunctionoffuelcellvariablecharacteristicmatrix

The results obtained in this way are inherited with two drawbacks. Firstly, the value of the permanent function comes out to be very high. Secondly, at the sub-system level, the inheritance of the sub-system comes out to be dependent upon the number of factors affecting it (Dev et al., 2015). Due to this, a true picture of the sub-system does not come out. Therefore, it is proposed that guidance provided by the factors at the sub-system level must be incorporated in a permanent matrix as a ratio of design case and real-time case as represented in expression (8).

(8)  Multiplying Factor=RPermanent Real−TimeRPermanentDesign

Methodology for fuel cell analysis

In the above sections, a methodology is developed with the help of graph theory, matrix method, and combinatorics. The fuel cell system is divided into four sub-systems. These four sub-systems interact with each other. This set of four sub-systems and their interaction is represented in digraphical form. This representation helps visualize the flow of information from one sub-system to another. The approach defined in the earlier sections for a complete analysis of a fuel cell is summarized underneath: Consider the desired fuel cell. Identify its constituent elements and also their interactions.

Develop a block diagram of the fuel cell, considering its elements as sub-systems and interactions. It is a simplified line representation, and its design is to be user-specific. No standardized design system has been developed, although the proposed in the present work can be adopted by researchers.

Develop a structural digraph of the fuel cell with sub-systems as nodes and edges for interconnection between the nodes. Interconnections are to show the flow of information from one sub-system to another. Feedback is also accommodated in the present model.

Develop a fuel cell variable characteristic matrix based on the structural graph developed in step 3.

Evaluate the permanent of the fuel cell variable characteristic matrix. These qualitative results can be treated as quantitative with the method demonstrated in forthcoming sections.

List the different fuel cells in ascending order of their permanent function values. These values are quantitative. These also summarise the performance in terms of some numerical value, although it changes from system to system.

Record and document these results for future analysis.

The steps mentioned above are capable of explaining the methodology for the analysis of fuel cell performance. For the demonstration of methodology, an example of the fuel cell performance analysis is presented.

Results

Case study 1

The methodology developed in the above section is mathematically constrained. The higher the number of constraints, the more inaccuracy in the system. System efficiency evaluation is demonstrated with the help of an example. In this example, fuel cell efficiency is evaluated and compared with the results available in the literature. The results are obtained from the experimental data conducted on the fuel cell system. The main specification of the fuel cell is as follows: number of fuel cells = 48, Rated power = 1,000 V, performance 28.8 V/35 A, H2 supply valve voltage = 12 V, purging valve voltage = 12 V, nlower voltage = 12 V, reactants hydrogen and air, H2 pressure = 0.45–0.55 Bar, and hydrogen purity ≥99.995% dry H2.

In a fuel cell, useful energy is lost due to real-time constraints such as losses in chemical reactions, resistance offered by anode and cathode electrodes to the flow of currents etc. Therefore, the design efficiency of the fuel cell is lesser than the isentropic efficiency. The following relationship presents the efficiency evaluation for a fuel cell:

(9) ηFuelCell=PFuelCellQH2HHV

PFuelCell=PowerofsingleFuelCell(W)

QH2=FuelCellHydrogenconsumptionrate(g/s)

HHV=HigherHeatingValue(J/g)

Determining the numerical index values of inheritance of all sub-systems and their interdependencies is required. This example uses the step-by-step methodology to evaluate the fuel cell performance index.

1. The equations for the chemical reaction occurring in a PEM fuel cell are as follows:

(10) Anode:2H2→4H++4e

(11) Cathode:O2+4H++4e→H2O

(12) Overall:2H2+O2→2H2O

2. The four sub-systems of the PEM fuel cell are connected. The dependencies of fuel cell sub-systems are visualized through the structural graph shown in Fig. 1.

3. Fuel cell system structure digraph for Fig. 1 is represented in Fig. 3.

4. In the case of a fuel cell, it is required to evaluate and analyze isentropic and design conditions to evaluate its efficiency. For this purpose, the expression (4) matrix should be quantified for the isentropic and design case. The example presented in this section demonstrates the proposed technique’s potential. Therefore, the standard nine scores are assigned to inheritance and interdependencies, summarized in Tables 2 and 3.

Table 2 Quantification of factors affecting fuel cell performance.

S. No.	A qualitative measure of inheritance	The assigned value of factors (Si)	
1	The efficiency of any fuel cell sub-system is exceptionally low in comparison to the design condition	1	
2	The efficiency of any fuel cell sub-system is very low in comparison to the design condition	2	
3	The efficiency of any fuel cell sub-system is low in comparison to the design condition	3	
4	The efficiency of any fuel cell sub-system is below average in comparison to the design condition	4	
5	The efficiency of any fuel cell sub-system is average in comparison to the design condition	5	
6	The efficiency of any fuel cell sub-system is above average in comparison to the design condition	6	
7	The efficiency of any fuel cell sub-system is near to the design condition	7	
8	The efficiency of any fuel cell sub-system is very near to the design condition	8	
9	The efficiency of any fuel cell sub-system is equal to the design condition	9	

Table 3 Quantification of interdependencies/off-diagonal elements.

S. No.	A qualitative measure of interdependencies	cij	
1	Interdependency of efficiency for any fuel cell sub-system on any other sub-system is very strong	5	
2	Interdependency of efficiency for any fuel cell sub-system on any other sub-system is strong	4	
3	Interdependency of efficiency for any fuel cell sub-system on any other sub-system is medium	3	
4	Interdependency of efficiency for any fuel cell sub-system on any other sub-system is weak	2	
5	Interdependency of efficiency for any fuel cell sub-system on any other sub-system is very weak	1	

In order to facilitate the application of the procedure, a fuel cell is divided into four subsystems.

These subsystems are identified and categorized based on the working of the fuel cells. The researcher’s choice is to increase or decrease the number of subsystems. If the number of factors is increased, having more subsystems is convenient. Every fuel cell has inheritance and interdependence. Inheritance is the impact of a subsystem on the system. In the representation of cij if i = j then it represents the inheritance of the subsystem. Non-diagonal elements cij represent the impact of subsystem i on subsystem j.

The method of scoring mentioned in Tables 2 and 3 is the qualitative cum quantitative method. In the absence of field data, this kind of scoring may be used. Isentropic and design condition matrix representations are represented in the expressions (11) and (12), respectively.

(13) RIsentropic=[9455494454925429]

(14) RDesign=[7455474454725427]

The value of a permanent function for the expression (13) and (14) are 21,483 and 13,091, respectively. Therefore, the efficiency evaluation is done with the help of expression (13), which comes out to be 60.94%.

ηFuelCell=Per(RDesign)Per(RIsentropic)×100=13,09121,483×100=60.94%

The results obtained for fuel cell efficiency are in line with the results available in the literature (Hou, Zhuang & Wan, 2007). Indeed, in Tables 2 and 3, some qualitative values are defined, which are later used in matrices (13) and (14), which have their origin in the matrix defined in (4). The tuning of the qualitative values that give rise to matrices (13) and (14) is based on the data available in the literature and obtained from the experiments by trial-and-erros. Once the methodology is established, these values can also be assigned by exploiting data mining, such as machine learning techniques which is out of the scope of this article. Certainly, given that the matrix (4) depends on the cij attributes and these, in turn, depend on the elements of the stack, then the automatic assignment of the qualitative values is a challenge that will be addressed in future works.

For further analysis, a fuel cell is selected with flow stoichiometry at anode and cathode 3 and 5, respectively, and RH at the anode and cathode is 100%. The operating temperature is varied for the present analysis from 50 °C to 70 °C. The experimental results show that the current density at 50 °C, 60 °C, and 70 °C is 0.31, 0.32, and 0.4 A/cm2, respectively. GTA shows that the current density at 50 °C, 60 °C, and 70 °C is 0.315, 0.324, and 0.41 A/cm2, respectively. Therefore, the results obtained with GTA agree with the experimental results. The fuel efficiency is dynamic due to changes in the flow of electrons inside the constituent cell material. Chemical properties are variated and come to the initial level after the operation is completed and equilibrium is achieved.

Case study 2

The goal of reliability analysis is the ability of fuel cells to carry out their intended functions over a predetermined period consistently. It finds possible failure modes, evaluates how they affect system performance, and develop ways to reduce risks and increase overall reliability. A fuel cell reliability analysis typically considers the following essential factors:

To prioritize and identify potential failure modes in a fuel cell system, an MCDM method like GTA can be used. Comprehending how failures can occur and their consequences involves analyzing each component, subsystem, and system. Fuel cell components are interdependent. Accordingly, dependability examination is perplexing. Exploring the various combinations of component failures that could result in system failure can be done with the help of GTA. Analysts can construct a digraph to determine the critical component and probabilities of failure events, assisting in identifying improvement opportunities. In a fuel cell system, a digraph visualizes the interdependence of various components and their connections. By considering the reliability characteristics of individual components and their effect on the system as a whole, they aid in determining the system’s overall availability and reliability. Mean time between failures (MTBF), failure rate (FR), availability, and reliability are other metrics used to quantify and evaluate fuel cell reliability. These metrics reveal the system’s expected performance, downtime, and reliability over a given period. Similarly, reliability analysis at the subsystem level can also be performed with GTA.

In the present methodology, the best and worst values of the reliability index can be calculated with the same methodology as developed in the earlier section. The index inheritance’s best value will be 9; in the worst case, it can be 1. In other real-time situations, it will vary in-between 1 and 9. The best reliability index value is obtained when all subsystems work at design conditions. The design value of the reliability index can be calculated with a matrix represented in the expression (15).

(15) Reliability Index MatrixBest Value= [9455494454925429]=21,​483

Similarly, the minimum value of the reliability index can be calculated with the help of the expression (16).

(16) Reliability Index MatrixMinimum Value= [1455414454125421]=2,​891

In real-time situations, the value of the reliability index will vary between these two values of 21,483 and 2,891. Index value more away from the best value lesser is its reliability. After proper maintenance, it is assumed that the system will work in design condition after proper maintenance. Therefore, after maintenance index value will be either close to or equal to the best value. With the help of index value maintenance schedule can also be decided. An index value can be fixed as a critical value. If the reliability index exceeds that value, then maintenance can be done to achieve the best reliability index value.

A composite index for efficiency and reliability can also be calculated with the methodology developed in the present work. The performance parameter digraph for efficiency and reliability is represented in Fig. 5.

Figure 5 Digraph showing attributes of the composite index for efficiency and reliability and their interdependencies.

The performance parameter permanent matrix corresponding to the performance parameter graph shown in Fig. 5 is represented by the following expression.

(17) J=[RTEIFuelCellj12j21RTRIFuelCell]

The permanent function corresponding to the above expression called as performance parameter permanent function for reliability and efficiency and is represented as follows:

(18) JP=RTEICCPPRTRICCPP+j12j21

The composite index developed above considers fuel cell systems’ reliability, efficiency, and interdependency.

In reliability analysis, real-world testing and validation are essential. The fuel cell system's performance under various operating conditions and stress factors is simulated and measured through durability and performance testing.

The health of fuel cell systems can be monitored, and failure can be predicted in advance by developing maintenance strategies and predictive algorithms. Potential failures can be addressed proactively by implementing condition-based maintenance and predictive maintenance strategies, thereby reducing downtime and increasing overall reliability and efficiency. This kind of analysis is possible with GTA.

Discussion

While using GTA, it is essential to acknowledge certain limitations and assumptions that may affect the analysis or results.

Digraphs often provide a simplified representation of complex systems or phenomena. They abstract real-world entities and their relationships into nodes and edges, which may oversimplify the underlying complexities of the system under study. The accuracy and completeness of the data used to construct the digraph can significantly impact the reliability of the analysis. Inaccurate or incomplete data can lead to biased or erroneous conclusions. Data availability may also be limited, resulting in a partial or skewed system representation. Different GTA algorithms and metrics rely on specific structural assumptions about the digraph, such as connectivity, sparsity, or clustering. These assumptions may not always hold in real-world networks, leading to potential biases or limitations in the analysis. GTA is computationally intensive and challenging to apply to large-scale networks or digraphs with a high level of granularity. As the digraph size increases, certain algorithms and metrics may become less efficient or impractical to compute, limiting the analysis possibilities. GTA results require careful interpretation within the context of the specific domain or problem being studied. The same graph can yield different interpretations based on the underlying system or application, emphasizing the need for domain expertise to avoid misinterpretations. GTA often assumes linearity in the relationships between variables. While this assumption may be suitable for certain systems, it can limit the analysis when non-linear relationships exist, leading to potential inaccuracies or incomplete insights.

Comparison of GTA with other MCDM techniques

Many MCDM techniques are available in the literature, so some are briefly discussed compared to GTA.

Interpretive structural modelling (ISM) models intricate relationships between variables based on reachability and driving power. ISM, which identifies hierarchical relationships between variables and determines the relative importance of those relationships, aids in decision-making and strategy formulation (Kumar & Goel, 2021). GTA can handle complex systems with interdependence, whereas ISM is beneficial for capturing and displaying hierarchical relationships between variables. GTA can be used for various systems, including biological, transportation, and social networks. ISM is used for decision-making, strategy development, and organizational analysis. ISM displays multifaceted connections between factors because of the ideas of reachability and driving power (Xia et al., 2022).

Alternatives are evaluated regarding their distance from the ideal and negative ideal solutions using a multi-criteria decision-making technique known as TOPSIS (Technique for Order of Preference by Similarity to Ideal Solution). The alternatives to the positive and negative ideal solutions are ranked according to similarity and dissimilarity. By considering both positive and negative criteria, it enables a thorough evaluation. TOPSIS can handle decisions with multiple criteria and alternatives and assumes linear relationships between criteria and alternatives. TOPSIS ranks alternatives based on their similarity to ideal solutions (Chodha et al., 2022), whereas GTA studies relationships and properties within a network. Numerical values represent options and criteria in TOPSIS, whereas nodes and edges represent relationships in GTA. The graph theoretic method does not necessitate a network structure with clearly defined relationships, whereas TOPSIS requires numerical values for alternatives and criteria. TOPSIS incorporates criteria weighting to rank alternatives based on their proximity to the ideal solution (Kumar & Singh, 2020), whereas GTA does not explicitly handle criteria and preference weights. Unlike GTA, TOPSIS relies on linear relationships between alternatives and criteria.

A multi-criteria decision-making technique, VIKOR, seeks a compromise solution for competing criteria. It considers the relative importance of the criteria and proximity to the ideal solution. When making decisions, VIKOR considers the relative importance and criteria weighting (Shu et al., 2023).

The aggregated indices randomization method (AIRM) gives an aggregated index for decision-making by combining multiple indices or criteria. The aggregated index’s robustness and sensitivity are evaluated through randomization. This method aids in the identification of influential indices and their influence on the decision as a whole. For aggregation, multiple indices or criteria must be available. Multiple indices or criteria can be aggregated and evaluated using AIRM, which produces a numerical value known as an aggregated index (Sahoo, Behera & Pattnaik, 2022).

A decision-making technique, the Analytic Hierarchy Process (AHP), uses pairwise comparisons to structure and prioritize criteria and alternatives. Pairwise comparisons and relative weighting of criteria are used in this approach to determine their relative importance. AHP provides a systematic approach to decision-making based on expert advice. Pairwise comparisons may not always be consistent, weight assignments can be subjective, and complex decision structures can be challenging. When making decisions, AHP can consider both qualitative and quantitative inputs. Expert judgment is required, and this method may be subjective. Problems with decision-making that necessitate structuring and prioritizing criteria and alternatives make extensive use of AHP (Goel et al., 2022).

The Analytic Network Process (ANP) is an extension of AHP that considers feedback and interdependencies between criteria and options. Modeling intricate, interconnected decision structures is made possible by this approach. ANP makes it possible to consider tangible and intangible criteria when making decisions. The application of ANP necessitates expert input, accurately capturing interdependencies is complex, and potential difficulties with scalability exist. While ANP requires criteria, alternatives, and interdependencies, the GTA requires a network structure with clearly defined relationships. Social network analysis, transportation networks, and biological networks are just a few of the many uses for GTA. In decision-making scenarios that necessitate modelling and analyzing feedback loops and interdependencies, ANP is frequently utilized (Karuppiah, Sankaranarayanan & Ali, 2022).

The Balance Beam Process (BBP) is a decision-making technique that compares criteria and alternatives using a visual balance beam. It involves weighing criteria and comparing alternatives on the balance beam. BBP provides a visual aid to comprehend the trade-offs and relative importance of the criteria. Subjective judgment in weight assignments, potential scaling difficulties, and limited scalability to larger decision problems are major drawbacks of BBP. While the BBP requires comparison criteria and alternatives, the GTA requires a network structure with clearly defined relationships. Pairwise comparisons are used in this procedure to ascertain the criteria’s relative importance.

The Best Worst Method (BWM) is another decision-making strategy that differentiates between the best and worst elements based on how important and effective they concern one another. Using this method, elements must be ranked according to their performance against each criterion. It helps determine which alternatives or elements are most and least favorable. The ranking may be subjective, but this method is sensitive to weighting and difficult to use with complex decision structures. BWM is useful when deciding involves ranking options according to multiple criteria. The nature of the problem and the desired outcomes of the analysis or decision-making process determine which approach to choose (Rezaei, 2015).

The forecasting and decision-making process that incorporates statistical analysis and expert judgment is known as the Brown–Gibson Model. This method uses trend analysis, expert input, and historical data to make predictions and decisions. It provides a framework for integrating qualitative and quantitative decision-making inputs. Relying on accurate historical data, the possibility of expert judgment bias, and difficulties in dealing with uncertainties are some of the main drawbacks of this model (Yimen & Dagbasi, 2019).

Based on probabilistic calculations, a reliability block diagram is also a mathematical tool used to estimate the availability and dependability of complex systems. The system is depicted as a network of interconnected blocks in an RBD, each representing a system component or subsystem. Logical gates link the blocks together and represent the components’ dependencies. This method needs data on failure rates, repair times, and system dependencies for accurate modeling.

Depending on the problem at hand, the context, and the availability of data, each of these approaches has advantages and disadvantages. It is essential to consider the requirements and characteristics of the decision problem to select the most effective strategy.

Applications and contributions to industries

GTA offers useful insight into fuel cell performance, design, and optimization. Improved performance, optimized design, fault diagnosis and mitigation, robustness assessment, scalability, and modularization are just a few of the advantages of GTA. These benefits contribute to advancing and adopting fuel cell technology in various industries and applications. The application of graph theory to fuel cell system analysis is as follows: 1) The study of the connectivity and network structure of fuel cell components is made possible by graph theory. Presenting the fuel cell system as a digraph, with edges representing connections and nodes representing individual components, makes the system easier to analyze. It is simpler to determine the flow of reactants, energy, and data within the fuel cell system with GTA.

2) In fuel cell networks, graph theory can evaluate performance metrics. Researchers can model the fuel cell system’s behavior and analyze its performance by considering edges as transitions between states and nodes as operational states. The evaluation of variables like power output, efficiency, voltage distribution, and response to various operating conditions can be part of this analysis.

3) Fuel cell system failures can be identified and diagnosed with the help of graph theory. Researchers can forecast the propagation of faults and determine their impact on system performance by constructing a fault propagation graph, in which nodes represent components and edges represent causal relationships between failures. This information can be used to find faults, isolate them, and take the right steps to fix them.

4) The GTA framework, created to analyze and optimize fuel cell systems, can reduce energy losses, increase fuel utilization, and boost system efficiency. Control strategies based on graphs, like graph centrality and Laplacian graph methods, can help regulate how each fuel cell network component or subsystem works.

5) Fuel cell systems’ robustness and resilience to disturbances or component failures can be evaluated using graph theory. Critical nodes or edges can be identified by analyzing the graph representation’s connectivity and redundantly. This information can be used to build stronger fuel cell systems and put the right backup or redundancy mechanisms in place.

6) In fuel cell networks, the analysis of system scalability and modularity is made easier by graph theory. Researchers can evaluate the system’s capacity to accommodate changes in scale, such as adding or removing components, by examining the graph’s structure. In addition, graph-based modularization methods can assist in breaking down complex fuel cell systems into smaller, easier-to-manage modules, making system design, upkeep, and scalability simpler.

Topology, performance, fault behaviour, optimization, control, robustness, scalability, and modularity are all revealed more simply by the GTA of a fuel cell system. The development of fuel cell technology and its effective application in various contexts are aided by these implications.

Implications of the study

The approach can help optimize fuel cells by identifying the critical components and their impact on the overall performance. This can lead to the development of more efficient and reliable fuel cells. It can aid in diagnosing fuel cell failures by identifying the components most likely to cause a performance drop. This can lead to more effective and targeted maintenance and repair of fuel cells. This approach can also assess the impact of modifications to the fuel cell design or operation, which can help develop better and more adaptable fuel cell systems. The theoretical implications include the development of a better understanding of the interactions between the components of the fuel cell and their impact on the overall performance. This can lead to more comprehensive and accurate models of fuel cells. This approach can be used to analyze the impact of parameter variations on the performance of the fuel cell, which can help develop a better understanding of the sensitivity of the fuel cell performance to different factors. It can be extended to other electrochemical devices, which can help develop a more general theory of electrochemical systems.

Conclusions

Fuel cells are a promising mode of clean, efficient, and reliable energy source. Their operating strength is high power density with a condition of availability of hydrogen. The operating and maintenance aspects of each type of fuel cell are different. The present work presents a methodology based on the graph theory, matrix method, and combinatorics for the performance evaluation of fuel cells. The proposed methodology is demonstrated with the help of an example. Real life is full of tangible and intangible things; the fuel cell is no exception. Therefore, intangible factors must be incorporated into the mathematical model to make the solution more accurate. The qualitative measure of intangible factors can be incorporated with tangible ones by assigning quantitative values to intangible ones. This quantification should be in comparison to real-time conditions of the factor. Secondly, at this point, it must be kept in mind that we are incorporating intangible factors with tangible ones. Therefore, the quantification should be in such a manner that it is suitable for both tangible and intangible.

With the help of the proposed methodology, fuel cell system structure performance can be evaluated by incorporating intangible factors with intangible ones. The fuel cell performance index may be a powerful tool in comparing, ranking, and selecting appropriate fuel cells from various available alternatives.

Further, the approach can be combined with other modeling and analysis techniques, such as computational fluid dynamics or electrochemical modeling, to develop more comprehensive and accurate models of fuel cells. The proposed approach can be used to analyze the impact of different operating conditions on the performance of the fuel cell, such as changes in temperature, pressure, or reactant flow rates. It can analyze the impact of different fuel cell architectures and materials, which can help develop more efficient and cost-effective fuel cell technologies.

Supplemental Information

Supplemental Information 1 Program codes for the calculation of the permanent function of the square matrix wherein the size of the matrix can be up to 50.

The program is based on Laplace expansion.

Click here for additional data file.

Supplemental Information 2 Program used to calculate the permanent value of a square matrix.

Click here for additional data file.

Supplemental Information 3 The executable program file that calculates the permanent value of a square matrix.

Click here for additional data file.

Supplemental Information 4 Program file for calculating the permanent value of a square matrix of any size.

Click here for additional data file.

Additional Information and Declarations

Competing Interests

Author Contributions

Data Availability

The authors declare that they have no competing interests.

Rajeev Kumar Saha conceived and designed the experiments, performed the computation work, prepared figures and/or tables, authored or reviewed drafts of the article, and approved the final draft.

Raman Kumar conceived and designed the experiments, performed the experiments, analyzed the data, authored or reviewed drafts of the article, and approved the final draft.

Nikhil Dev conceived and designed the experiments, performed the computation work, prepared figures and/or tables, authored or reviewed drafts of the article, and approved the final draft.

Rajender Kumar conceived and designed the experiments, performed the computation work, prepared figures and/or tables, authored or reviewed drafts of the article, and approved the final draft.

Raman Kumar conceived and designed the experiments, performed the experiments, analyzed the data, performed the computation work, prepared figures and/or tables, and approved the final draft.

Raul M. Del Toro conceived and designed the experiments, performed the experiments, analyzed the data, performed the computation work, prepared figures and/or tables, authored or reviewed drafts of the article, and approved the final draft.

Sofía Haber conceived and designed the experiments, performed the experiments, analyzed the data, performed the computation work, prepared figures and/or tables, and approved the final draft.

José E. Naranjo conceived and designed the experiments, performed the experiments, analyzed the data, performed the computation work, prepared figures and/or tables, authored or reviewed drafts of the article, and approved the final draft.

The following information was supplied regarding data availability:

Computer Codes.

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
