# Peer review of "Structural modeling and analysis of fuel cell: a graph-theoretic approach"

_PeerJ Computer Science, doi:10.7717/peerj-cs.1510_

## Round 0.1 · original submission · Minor Revisions

The reviewers have some concerns about this manuscript. The authors should provide point-to-point responses to address all the concerns and provide a revised manuscript with the revised parts being marked in different color.

Reviewer 1 ·

Basic reporting

Overall the paper is well written, figures, tables of the paper are carefully designed.

Experimental design

The experiment is well designed and the author describes the experiment in a good manner. The paper presents a methodology for analyzing fuel cell system structure using graph theory and matrix method well. The methodology consists of a series of steps, including digraph representation, matrix representation, and permanent function representation. The fuel cell system is divided into four interdependent sub-systems, and a mathematical model is evaluated quantitatively to produce a performance index numerical value.

Validity of the findings

The paper only use one example as case study. I think one area of improvement could be to include more experimental data or case studies to support the proposed methodology, provide a more detailed explanation of the limitations and assumptions of the proposed approach. I believe providing at lease one more cases study using the proposed approach and comparing with other approaches can further strength the conclusions of the paper.

Cite this review as

Reviewer 2 ·

Basic reporting

1. The paper demonstrates a commendable level of clarity and professionalism in its English writing. However, it is advisable for the author to conduct a thorough review of the article to address minor errors that may have been overlooked. For instance, specific attention should be given to lines 52 to 55, which require rewriting and integration into the introduction section. By refining these areas, the paper will achieve a greater cohesiveness and improve the overall flow of information.

2. It is crucial to ensure consistency in the formatting of equations throughout the paper. It is recommended to carefully verify all equations and confirm that they adhere to the same format. By doing so, the paper will maintain a uniform style and facilitate a smoother reading experience for the audience.

Experimental design

3. In my opinion, repositioning the "Demonstration of Proposed Methodology" section as the "Results" section would enhance the overall understanding of the paper for readers. Presenting the findings of the proposed methodology in the results section would provide a more logical and intuitive structure. This modification would allow readers to readily comprehend the outcomes of the study and their significance within the context of the research.

Validity of the findings

It is important to incorporate a comprehensive discussion section that effectively addresses the implications of the study. The authors should consider adding a dedicated discussion section to the paper. This section should encompass several key aspects, including highlighting the strengths of the proposed methods, comparing them with other existing methods, identifying the specific problem(s) they aim to solve, discussing the broader implications of the study's findings, and acknowledging any limitations encountered during the research process. By incorporating these elements, the discussion section will contribute to a more insightful and well-rounded interpretation of the study's results.

Additional comments

The figures presented in the paper would benefit from improvements in terms of their quality and presentation. It is advisable to enhance the visual clarity and labeling of the figures to ensure they effectively convey the intended information to readers. By refining the figures, the paper will enhance its overall visual appeal and facilitate a better understanding of the data and concepts being presented.

Cite this review as

---

## Round 0.2 · accepted · Accept

Reviewers are satisfied with the revisions made in this version and I would recommend accepting this manuscript.

Reviewer 1 ·

Basic reporting

N/A

Experimental design

N/A

Validity of the findings

N/A

Additional comments

The revision addressed all the comments. no further feedbacks.

Cite this review as

Reviewer 2 ·

Basic reporting

Good

Experimental design

Good

Validity of the findings

Good

Cite this review as